# Over the counter use of topical corticosteroid for skin conditions among patients before attending skin specialist clinic in Nepal: A qualitative study

Yogesh Poudyal[1*], Nishika Aryal[2], Anuja Rajbhandari[2], Mukti Ghimire[3], Sanjaya Acharya[4], Shristi Raut[5], Bishal Joshi[6], Sunil Pokharel[7], Phaik Yeong Cheah[7,8], Bipin Adhikari[7,8,9]

1 Sarnath Skin Centre, Shanti Path, Bhairahawa, Nepal, 2 Department of Public Health, La Trobe University, Melbourne, Australia, 3 Department of Pediatrics, Nepal Police Hospital, Krishnadhara Marg-3, Kathmandu, Nepal, 4 Academy for Data Science and Global Health, Kathmandu, Nepal, 5 Department of Microbiology, Institute of Medicine, Tribhuvan University, Maharajgunj Campus, Kathmandu, Nepal, 6 Department of Physiology, Universal College of Medical Sciences, Bhairahawa, Nepal, 7 Centre for Tropical Medicine and Global Health, Nuffield Department of Medicine, University of Oxford, Oxford, United Kingdom, 8 Mahidol-Oxford Tropical Medicine Research Unit, Faculty of Tropical Medicine, Mahidol University, Bangkok, Thailand, 9 Central Department of Microbiology, Tribhuvan University, Kirtipur, Nepal

* neverland791@yahoo.com

## Abstract

Access to topical corticosteroids (TCs) globally includes over the counter (OTC) sale without proper guidelines, exacerbating treatment-resistant infections and complicating outcome of skin conditions. The main aim of the study was to evaluate the reasons behind OTC use of TCs among patients with skin conditions before attending a skin specialist clinic. This study was conducted from September 2021 to March 2022. The study included in-depth interviews (IDIs) and observations, among a total of thirty-one patients with specific skin conditions using TCs, at the Sarnath Skin Centre in Siddhartha Nagar (Bhairahawa), Nepal. All interviews were conducted in the Nepali language, audio-recorded, transcribed and translated into English and were coded and analyzed in NVivo. Among the 31 individuals diagnosed with Dermatophytosis, Melasma, Eczema, common treatments included use of Sonaderm (Clobetasol, Gentamycin and Miconazole), Derma-KT (Clobetasol, Gentamicin, Ketoconazole and Iodochlorhydroxyquinoline) and steroid-infused products from non-medical sources for prolonged periods of time, exposing patients to transient recovery and perpetuating a vicious cycle of OTC treatment seeking until patients failed to recover. OTC seeking was facilitated by participants' easy access to drug dispensers, suggestions from and practice of family and friends which often led to temporary relief for a few weeks/months. In addition, access factors (e.g., distance, cost, time), convenience factors (e.g., ease of presenting at the drug store compared to skin clinic) also influenced the choice for OTC. There was a lack of knowledge related to adverse effects of topical steroids, including proper guidance on the use of topical medications and limitations of home remedies. OTC use of topical medications poses significant challenges, often leading

**Data availability statement:** Data cannot be shared publicly because of the risk that respondents could be identified. Following the ethical guidelines of Nepal Health Research Council on data sharing policy and written informed consent with participants where we ensured the anonymity and confidentiality of the information provided to us, data cannot be shared publicly. Even when sharing the data after removing the socio-demographics of the participants, their specific skin conditions (sensitive information), and the dates of study they took part in are likely to reveal their identification. However, data are available on reasonable request to Nepal Health Research Council (E-mail: nhrc@nhrc.gov.np).

**Funding:** The study received funding from Society of Dermatologists, Venereologists and Leprologists of Nepal (SODVELON) only adequate for the data collection. SODVELON is an umbrella organization established for the welfare of dermatologists of Nepal. The study is also part of the foundational work supported by Global Health Bioethics Network (GHBN), UK (228141/Z/23/Z). BA is a GHBN fellow. For Open Access, the author has applied for a CC BY public copyright license to any Author-Accepted Manuscript version arising from this submission. The funder had no role in the writing or preparation of the manuscript.

**Competing interests:** The authors have declared that no competing interests exist.

to complicated skin cases that present late at skin-specialist clinics. A multi-pronged public and community engagement approach is critical to curb the treatment seeking dynamics among patients developing skin conditions.

## Introduction

Topical corticosteroids (TCs) have been a cornerstone of dermatologic therapy since the 1950s and are widely prescribed by dermatologists in outpatient care [1]. Available in various forms—creams, ointments, gels, and solutions—TCs are among the most frequently prescribed dermatologic medications [2,3]. However, their widespread use has become a double-edged sword. Many are misused for conditions such as pigmentation, acne, pruritus, infections, and rashes due to their lightening and anti-inflammatory properties [4].

TC abuse is particularly prevalent in Asian and African countries, where colorism and a preference for lighter skin have deep social implications [5]. Contributing factors include inadequate health infrastructure, limited access to specialists, self-medication practices, easy over-the-counter (OTC) availability, incorrect prescriptions, aggressive pharmaceutical marketing, and poor regulation of irrational drug combinations [6,7].

In Nepal, the OTC use of antimicrobials remains a significant issue. Studies have identified regulatory gaps and highlighted increased OTC purchases, more so during the COVID-19 pandemic [8–10]. Ineffective oversight of medicine shops enables unrestricted drug sales [11]. General practitioners and pharmacists often recommend such products due to their potential for quick symptom relief [12]. In 2013, the top-selling topical steroid combination—beclomethasone, neomycin, and clotrimazole—generated Rs. 152 crores (approx. USD 25 million), despite lacking scientific rationale [4,13].

Many topical preparations in Nepal are sold OTC for prices ranging from USD 0.5 to 4, often without a prescription, and in the absence of clear guidelines distinguishing OTC and prescription-only medicines [14]. Dermatophytosis, also known as ringworm, is a common superficial fungal infection affecting the skin, hair, and nails [15]. Dermatophytosis is ranked among the top four disease burdens globally [16]. Among skin conditions, dermatophytosis poses a significant shift in clinical profile, with rise in chronic, recurrent, or treatment resistant cases [17]. The American Academy of Dermatology (AAD) and the International League of Dermatological Societies (ILDS) are tracking such cases through the "AAD/ILDS Dermatology COVID-19, Monkeypox, and Emerging Infections Registry" [18]. The OTC use of topical steroids has been identified as a contributing factor to treatment-resistant tinea [19]. Irrational combination creams—containing potent steroids, antifungals, and antibacterials—are widely available and are considered major contributors to this alarming trend [19,20].

In Nepal, most of the population resides in remote areas where healthcare is provided by health assistants and general practitioners in primary health care centers and district hospitals [21]. Although the Ministry of Health offers basic healthcare

services for free, the rural health services remain limited [21,22]. Specialist care, including dermatology, is largely confined to hospitals in major cities or medical colleges [22]. Many health workers lack adequate dermatological training, leaving large segments of the population without proper skin care services [23–25]. Consequently, dermatology clinics in Nepal frequently manage steroid-induced or steroid-aggravated dermatoses [25,26] which can be more difficult and distressing to treat than the original condition [27]. In more than 93% of the skin cases, TCs are used unnecessarily—often for extended durations, at unsuitable potencies, or without a proper diagnosis [11].

While previous studies have documented the burden of topical steroid misuse [4,12,27–29], few have explored patients' perspectives, particularly the lived experience of skin condition and treatment seeking behaviors. The main objective of this study was to explore the patient's experience of dermatological conditions, the treatment seeking behavior including decision to use OTC steroid topical preparation and the consequences.

## Methods

### Ethics statement

This study received ethical approval from Nepal Health Research Council (NHRC Ref # 388, August 27, 2021). All respondents provided written informed consent to participate in the study.

### Design and overview

This was a phenomenological study that explored a particular phenomenon under the research question to explore the experiences and responses to it. The particular phenomenon in this study refers to the skin condition patients have developed and sought treatment at the dermatological clinic served by the first author (YP). To explore the phenomenon under the study, data were collected using in-depth interviews (IDIs) and observations of patients with skin conditions. The specific type of patients (respondents) for this study comprised patients with any skin diseases that made them use topical steroids. Grounded in the philosophical tradition of phenomenology, this study aimed to understand participants' lived experiences and their encounters with steroid use and skin conditions. The study followed a COREQ (Consolidated criteria for Reporting Qualitative research) guideline (S1 Text).

### Study setting

This study was conducted at Sarnath Skin Centre, a private dermatology clinic in the urban center of Bhairahawa. Established in 2017, the clinic serves approximately 40 patients daily, primarily those with skin-related conditions. Consultation fees at such clinics in the area typically range from USD 5–7.

Bhairahawa, officially known as Siddharthanagar, is a municipality and the administrative center of Rupandehi District in Lumbini Province, Nepal. Located on the southern plains near the Indian border, the city has a population of approximately 75,000 [30]. The city has a tertiary hospital hosted by Universal College of Medical Sciences with specialist clinics including dermatology. There are private hospitals and clinics where patients also seek specialist care.

Bhairahawa as an administrative and urban center of the district receives patients from the settlements around the urban center including patients from northern mountains who visit for specialist clinic care. Most of the patients attending the specialist clinic came from outskirts of the urban Bhairahawa and were mostly working in the farmlands, particularly rice fields.

### Selection of respondents

All respondents were selected purposively in this study, and the total number of participants was decided based on data saturation [31]. The data saturation in this study was considered achieved when additional interviews began to yield repeated themes, insights, nuances, and understandings of the phenomenon [32]. Patients with specific skin

conditions with prior history of use of topical corticosteroid were selected to align with the research question of this study. Participants were first verbally asked if they were interested in participating in the study during their clinical examination. Each prospective participant was briefed on the study, its procedures, and its implications, as outlined in the informed consent form. It was explained that their anonymized perspectives would contribute to the study findings, and the only potential risk—that they would need to share details about their skin conditions and the treatments they had sought—was clearly communicated. Once they verbally agreed, a written informed consent was obtained and were interviewed in the separate room within the clinic premise. Confidentiality was maintained by conducting interviews in private within the clinic, securely storing data on the first author's password-protected laptop, sharing data only among co-authors for analysis and interpretation, anonymizing the data when reporting, and removing identifiers wherever possible.

No repeat interviews were conducted, and each interview lasted around 45 minutes to 1 hour. The study was conducted between September 2021 and March 2022. Participants were also selected based on their logistical feasibility to sit for interviews after the clinical services. A total of 31 patients were interviewed including ten non-participant observations based at the drug shop. Non-participant observation notes were written by the first author on skin patients' interactions with the dispenser and their decision-making dynamics before they attended the dermatologist.

YP is a dermatologist (MBBS, MD) with more than 10 years of dermatological practice in the region. The interviews were conducted by YP with participants meeting following inclusion criteria: any participant aged 18 years or older who had experienced a long-standing skin condition that failed to recover after home-based treatments or treatment elsewhere. Participants were excluded if they were unwilling to allocate time for the interviews, had long-standing skin conditions but were seeking treatment for the first time at the clinic, were unsure about past treatments received, or had co-morbid conditions requiring concomitant treatments. In addition, individuals who were not physically or mentally fit, or unwilling to provide written informed consent were respected and excluded from the study.

### Data collection guide

Data were collected using an IDI guide that included a range of topics incorporating patients' experiences regarding their skin conditions, normative health-seeking behavior, knowledge of OTC and TC, and policies related to TC (S2 Text). The thematic guide for the interviews was prepared and reviewed by the authors (YP and BA). All interviews were conducted in Nepali and recorded using an audio recorder, with additional non-verbal cues and relevant information noted during the interviews and documented as field notes. Subsequently, all recorded interviews were translated and transcribed into English for analysis.

### Data analysis

Participants' personal information was anonymized to protect their identity. All transcribed data were stored in Microsoft Word and then analyzed by two coders (NA and AR) using NVivo software. Six phases of thematic analysis outlined by Braun and Clarke [33] were utilized that included 1. Familiarizing with the data (transcribing data, reading and rereading the data, noting down initial ideas); 2. Generating initial codes (coding interesting features of the data systematically across the entire data set, collating data relevant to each code); 3. Searching for themes (collating codes into potential themes, gathering all data relevant to each potential theme); 4. Reviewing themes (checking if the themes work in relation to the coded extracts and the entire data set, generating a thematic map); 5. Defining and naming themes (ongoing analysis to refine the specifics of each theme, and the overall story the analysis); and 6. Producing the report (selection of vivid, compelling extract examples, final analysis of selected extracts, relating back of the analysis to the research question and literature, producing a scholarly report of the analysis) [33,34].

The thematic analysis in this study entailed a mix of deductive and inductive approaches to coding of data and was critically reviewed by BA, YP and BJ. The major themes were informed by a framework of parent and sub-codes (S1 Fig).

The analyzed data were thoroughly cross-checked for any mis-association and misinterpretations. By focusing on the subjective experiences of the participants, the research sought to uncover deeper meanings and insights into their personal events.

### Reflexivity

YP, the principal investigator and treating dermatologist, conducted the interviews with patients (S3 Text). As a dermatologist, YP's interactions with patients may have been influenced by epistemic and power dynamics, potentially introducing social desirability bias in the responses.

NA and AR, both Nepalese public health practitioners, independently coded the data and generated the initial codes and themes. YP and BA participated regularly in the coding and thematic synthesis processes. BA is a social scientist and Nepali medical practitioner with prior research experience on the over-the-counter use of antimicrobials in Nepal and provided input on coding and the final thematic synthesis.

MG, SA, SR, BJ, and SP are medical doctors with extensive experience in Nepal's healthcare landscape. PYC is a professor of ethics with significant collaborative research experience on antimicrobial resistance in South and Southeast Asia.

## Results

The socio-demographic details of the participants are presented in Table 1. Most of the participants lived within a radius of 20 km from clinic site. Four broad themes were identified that included a) skin conditions and the participants knowledge, b) knowledge gap; and c) treatment seeking behavior; d) patient perception (**Fig 1**). These four broad themes were further consolidated into two major themes that included 1. Skin conditions and the knowledge gaps; and 2. Treatment seeking behavior.

### Skin conditions and the knowledge gaps

*Daaz, Daad* [dermatophytosis] was the most common skin condition reported by participants later confirmed by the dermatologist. Other frequently encountered skin issues in their locality were melasma, scabies, vitiligo, and acne. These conditions represented a significant portion of the dermatological problems faced by the community.

> *I don't exactly know the name, but in our language, we call it "Daad". Only Daad is prevalent in our family members and locality. – Pt-22, 20–30 years, male.*

Most of the participants associated their skin condition with factors such as hot weather, sun exposure, and sweating. They believed that these environmental and personal habits were primary causes of their skin conditions. Their overall basic knowledge about skin diseases was very limited. Even for diseases like dermatophytosis, which affected almost all the members of the household, they did not realize its infective nature.

> *Skin conditions are due to the use of wet clothes, wet underwear and due to sweat and if we don't bathe regularly. – Pt-16, 18–20 years, male*

People did not have sufficient knowledge regarding their skin diseases and its treatment processes. Whenever they were affected by any condition, they were unaware of where to go or whom to consult for healthcare services. It was common thinking among the participants that their disease was not complicated, unaware of its potential risks and resorting to haphazard drug use rather than seeking treatment from a certified specialist. But after prolonged illness and recurrent symptoms, they realized that the disease was not as common as they had initially thought.

**Table 1. Socio-demographics of the participants in the study.**

| S. N. | Age | Sex | Educational level | Profession | Diagnosis |
|---|---|---|---|---|---|
| 1 | 40–50 | F | Grade 10 | Housewife | Dermatophytosis |
| 2 | 20–30 | M | Grade 10 | Job | Dermatophytosis |
| 3 | 20–30 | F | Grade 10 | Housewife | Dermatophytosis |
| 4 | 30–40 | M | Grade 10 | Business | Dermatophytosis |
| 5 | 18–20 | M | +2 Level | Student | Dermatophytosis |
| 6 | 20–30 | F | Grade 9 | Farmer | Dermatophytosis |
| 7 | 20–30 | F | Grade 3 | Housewife | Dermatophytosis |
| 8 | 18–20 | M | Grade 10 | Student | Dermatophytosis |
| 9 | 30–40 | F | +2 Level | Job | Dermatophytosis |
| 10 | 40–50 | M | Grade 10 | Job | Dermatophytosis |
| 11 | 20–30 | F | Bachelor level | Student | Dermatophytosis |
| 12 | 30–40 | M | Grade 6 | Job | Dermatophytosis |
| 13 | 20–30 | M | Bachelor level | Job | Dermatophytosis |
| 14 | 20–30 | M | Bachelor level | Job | Dermatophytosis |
| 15 | 20–30 | F | Grade 8 | Housewife | Melasma |
| 16 | 18–20 | M | Grade 10 | Student | Dermatophytosis |
| 17 | 18–20 | M | +2 Level | Student | Dermatophytosis |
| 18 | 70–80 | M | Grade 10 | Business | Dermatophytosis |
| 19 | 18–20 | M | +2 Level | Student | Dermatophytosis |
| 20 | 50–60 | M | Diploma | Job | Dermatophytosis |
| 21 | 20–30 | M | Grade 10 | Farmer | Dermatophytosis |
| 22 | 20–30 | M | +2 Level | Businessman | Dermatophytosis |
| 23 | 18–20 | M | +2 Level | Student | Dermatophytosis |
| 24 | 20–30 | M | Master level | Student | Eczema |
| 25 | 20–30 | M | Grade 10 | Farmer | Eczema |
| 26 | 30–40 | M | Grade 10 | Job | Dermatophytosis |
| 27 | 40–50 | F | Grade 10 | Job | Dermatophytosis |
| 28 | 20–30 | F | Grade 10 | Housewife | Melasma |
| 29 | 30–40 | F | +2 Level | Housewife | Melasma |
| 30 | 30–40 | M | Grade 10 | Businessman | Dermatophytosis |
| 31 | 40–50 | M | Bachelor level | Job | Dermatophytosis |

F: Female, M: Male, most common variant of dermatophytosis was T. incognito

Job: Working in a small business firm or private institute with a fixed monthly salary.

Dermatophytosis: A superficial fungal infection of the skin, hair, and nails caused by dermatophytes.

Eczema: A skin condition characterized by varying intensities of itching and soreness, with signs including dryness, redness, excoriation, exudation, and fissuring.

Melasma: A condition manifested as hyperpigmented patches on the face that become more pronounced after sun exposure.

> ya at first, I used to think it simple…….in initial phase it used to get cured after applying ringguard …… so I took it simply……but now when it kept on repeating for so many years I thought of getting consultation. – Pt-31, 40-50 years, male

Due to this misinformation and knowledge gap, people practiced some unusual home remedies, such as using goat's urine, Harpic (a toilet cleaner containing hydrochloric acid), detergent powder, and even considered invocation rituals which was extremely concerning.

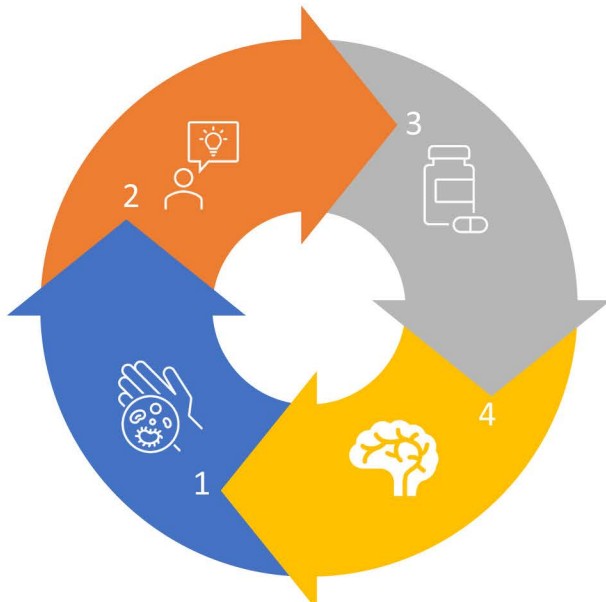

**2 Knowledge gap on the need for treatment**

- Poor knowledge about the skin conditions, their cause and potentials of transmission
- Poor knowledge on how to deal with such skin conditions, and the treatment process
- Lack of knowledge on how these skin conditions could evolve, and complicate
- Perceived lack of severity of the skin conditions

**1 Common skin conditions and the knowledge gaps**

- Dermatophytosis (*Daad; Daaj* were most common skin conditions
- Melasma, scabies, vitiligo, and acne were other common conditions
- Patients did not know the severity of these skin conditions, and often thought were caused by weather, sun exposure and sweating

**3 Treatment seeking behaviour**

- Patients first sought over the counter(OTC) medication including advice from family, friends, and neighbors
- Application of wide spectrum of home remedies
- OTC ointments often included steroids that suppressed the conditions for some time
- Inadequate recovery, or recurrence prompted them to seek treatment at the formal health services

**4 Perceptions on the skin condition and treatment**

- Patients perceived the cost of treatment at formal health services to be expensive
- Patients pronounced barriers to reaching formal health services (distance, the fact that they have to leave their work, travel arrangements)
- Stigma attached to skin conditions made some patients to conceal their condition deterring them to visit specialists.

**Fig 1. Major themes and some of the prominent findings.**

*Some people even suggested using goat urine, but that didn't work either. Then there were recommendations to use bitter gourd and camphor, so I tried mixing those as well, but nothing worked. Eventually, I became tired of applying anything at all. – Pt- 30, 30–40 years, Male.*

Patients were not properly instructed by the medical shop owner about the necessity of a prescription for medications. Another common phenomenon was that the medical shop owner/pharmacist/dispenser would normally provide medicines to the participants after they requested it, without inspecting and physically seeing their skin disease.

*….He used to ask what happened…..and I used to say "daaz"….than I used to ask for ringguard,.. and he used to give it immediately. – Pt-31, 40-50 years, male*

### Treatment seeking behavior

Observations at the skin clinic's pharmacy revealed that patients often wavered between consulting the specialist or trying over-the-counter medications first. Positive counseling from the dispenser about the benefits of seeing a dermatologist influenced many of their decisions. However, some patients opted to see the specialist without needing further discussion. Some of these patients who had already decided to see the skin specialist were either suffering from long-standing skin conditions or had not been cured by their repeated use of OTC medications.

Initially, when individuals noticed symptoms of their skin condition, they either went to a medical shop to get medication without a prescription or relied on advice from family, friends, neighbors, and relatives. They even applied home remedies or medications in the form of ointments, creams, and gels, which temporarily eased their symptoms, allowing them to return to their usual activities. However, after a few weeks or months, the condition recurred prompting them to obtain

the same treatment from the pharmacy. This cycle was repeated until they either became exhausted or their condition worsened significantly. This pattern illustrated their coping mechanism of relying on dispensers (pharmacists) for symptom management whenever their skin issues reappeared. Moreover, it was common for people in local communities to heavily rely on medical shop owners for medical advice and treatment, despite the lack of formal medical training. This habit underscored the accessibility and convenience of medical shops, which often served as the first point of contact in healthcare for many individuals.

*I got advice from the villagers who had suffered from similar disease earlier regarding how their disease got cured. I followed their advice. Then I thought, if I didn't cure then I visited a skin doctor. – Pt-19, 18–20 years, male*

*While using the medicine my face became very dry, it removed the black spots, but it looked as if the moisture from my face had disappeared. So, I stopped using that medicine. After that, I went again for an incantation (to invoke a magical effect or influence) through which I got some relief. I used aloe vera for 10–20 days, but nothing happened. Later on, one person gave me the address of this clinic. – Pt-29, 30–40 years, male*

Commonly used medicines for skin condition among participants included Sonaderm (clobetasol propionate, gentamycin sulphate, miconazole nitrate), Derma-KT (ketoconazole, neomycin sulphate, Iodochlorohydroxyquinoline, tolnaftate, clobetasol propionate), Derma-5 (clobetasol propionate, gentamycin, iodochlohydroxyquinoline, ketoconazole), Mofur (mometasone), quadriderm (clotrimazole, neomycin, beclomethasone), Melalite (hydroquinone, mometasone, tretinoin), Ring Guard (miconazole nitrate, neomycin sulphate, chlorocerol), SkinLite (Hydroquinone, mometasone, tretinoin). SkinLite was also found in non-medical settings such as fancy or cosmetic shops. It seemed they were used as cosmetics without understanding its potential side effects. Almost all of the participants were unaware of its composition and its potential side effects.

*Everything is available in the fancy shops. In Mahilwar, every shop has Chinese cream, and night cream like Melalite, and SkinLlite. I have a brother who owns a medical store. He gave me SkinLite and told me, "Sister, use this cream and you'll glow like a foreigner" and I used that cream and my skin became fair. But my skin was damaged with time. I used that for about 2–3 months. – Pt-15, 20–30 years, female*

Patients delayed seeking specialist care for their skin conditions due to various factors, such as the distance, time required to reach healthcare facilities, financial constraints, and the temporary relief provided by medications obtained from pharmacies. The long and significant travel distance to the clinic posed substantial difficulties for both first-time patients and those requiring regular visits. Most people relied on local transportation and bikes to reach the clinic, adding to the challenge of accessing necessary medical care.

*Today I also came all the way from Rajahar (approx. 80 km from the interview site) because I have some work over here and relatives here told me to get a checkup from you. – Pt-4, 30–40 years, male*

*I used medicine from India for six months, I used to travel to India every 15 days. – Pt- 29, 30–40 years, female*

Additionally, some patients were unaware of where to find dermatologists, while others felt embarrassed about their condition. However, some individuals prioritized finding an effective cure, even if the expenses were high, as they were tired of the recurrent nature of their skin problems. In the absence of professional medical guidance, patients often resorted to alternative remedies like turmeric, garlic, neem, aloe vera, camphor, alum, herbal juices, and bean leaves, which provided some temporary relief but did not address their underlying cause. Some of them also watched various advertisements on TV and the internet and tried the remedies/medications suggested for their problem.

*When I first noticed this disease, I thought it was just a simple irritation of skin folds and I applied a powder for 2–3 days. But nothing changed and that area became red. There were small red dots, and I thought about what to do with it. Then, one friend told me it is a simple daaz. I remember watching ring guard advertisements on TV. I searched for it, bought it and applied. – Pt-31, 40–50 years, male*

The excessive and prolonged use of TCs raised concerns, including diminished efficacy due to tachyphylaxis, skin atrophy, and steroid-induced dermatoses. Patients, often in an attempt to manage chronic dermatological conditions, resorted to applying TC creams and ointments far beyond recommended dosages and durations.

*I kept on applying those tubes 7–8 times after an interval of some months that is around 25–30 tubes for the whole 2 years. – Pt-26, 30–40, male*

Despite receiving immediate healthcare from specialist doctors on several occasions with multiple consultations, the patient continued to experience recurring issues with their condition. This indicated a more prevalent recalcitrant type of dermatophytosis, which is not even cured with the standard treatment regimen, making treatment long and costly. This has led patients lose faith in specialist consultation and has fueled steroid abuse.

*In the past two years, I went to the medical college and underwent some tests, but my condition still did not improve. I visited the doctor 2–3 times for regular follow-ups, and my condition completely disappeared, making me feel that the disease was gone. However, when it reappeared, I started to think that either the medication was incorrect or the doctor prescribed it without proper evaluation. I believed that even consulting a doctor wouldn't help, so I continued taking the same medication repeatedly. – Pt-3, 20–30 years, female*

Participants perceived that visiting a medical shop was more convenient, cheaper, and less time-consuming due to the lack of access to nearby doctors or information on where to find certified medical professionals. People who worked in medical shops and provided medications were informally referred to as "doctors," despite not necessarily holding medical degrees or qualifications. Many were unaware of the benefits of being treated by a certified doctor and often could not allocate time for a proper medical visit. Consequently, they adapted to using medicines from nearby medical shops, experiencing only temporary relief rather than a cure.

Participants also tended to hide their skin conditions out of fear of social rejection and shame. However, after suffering for extended periods, some were persuaded by close relatives to visit a specialist, particularly those who themselves had been successfully treated by a dermatologist. For others, unbearable pain eventually compelled them to seek professional help. Upon visiting a specialist, they realized they had been using unnecessary medications and ointments and discovered that their conditions could be permanently cured with appropriate treatment.

*We refer to the individuals who work in medical shops and dispense medications as "doctors". – Pt-23, 18–20 years, male.*

*They might feel ashamed because daaz often appears in areas like the groin. – Pt- 24, 20–30 years, male*

## Discussion

The study explored the OTC use of topical corticosteroids among patients attending outpatient clinics in Bhairahawa, with dermatophytosis being the most common issue alongside other conditions such as melasma, and Acne. Participants often associated environmental factors and inadequate hygiene practices as the causes of these conditions, leading them to frequently resort to self-medication. The delay in seeking specialist care was influenced by various factors including

distance, cost, and lack of awareness about proper treatment. Misuse of medications containing TC compounds was also observed, contributing to the side effects and prolonged suffering. The knowledge gap regarding skin diseases and their treatment processes was evident. Despite efforts to address these issues, challenges persisted due to a lack of steroid awareness, improper medication guidance, and a preference for convenience over specialist care.

Most cases of steroid misuse noted in this study were dermatophytosis. Tinea incognito and acne were the most frequent complications noted in patients misusing steroids in India [6]. A study by Molly et al [12] showed that 58% of the participants perceived their skin condition to be allergic to food when in fact 70.1% had Tinea. A number of studies from India, a regional neighbor, report high use of topical corticosteroids among dermatology patients and continue to highlight this as a major public health concern [3,4]. A recent review also demonstrated the widespread use of topical corticosteroids (TCs) within the Indian subcontinent, identifying it as a significant issue encountered in dermatology clinics [35]. The use of topical corticosteroids beyond the Indian subcontinent—such as in Southeast Asia—has similarly been reported as a major public health concern [36]. The over-the-counter (OTC) use of TCs reflects broader trends seen with other medications both within and outside the region, particularly in low- and middle-income countries (LMICs) [37]. Close to our study site, a study conducted in Bhairahawa showed that those who consulted a dermatologist were less likely to misuse steroids [38]. Our study echoed the poor knowledge of skin diseases among participants.

Management of skin diseases among our participants often seemed to start with home remedies, followed by seeking advice from their family circle and friends. If these early treatments failed, they seemed to seek advice and medicine from a nearby pharmacy. After long duration of trial and error with the OTC steroid often without recovery, or worse, when they developed complications, they finally sought treatment at the specialist clinics. A study by Saraswat et al [11] showed that more than half of patients received recommendations to use TCs from non-physician source such as peers or relatives, and drug dispensers. Similarly, drug dispensers and local healers contributed to 78% of the OTC use of steroids. Relying on non-physician source for the treatment was one of the important causes of OTC use of steroid.

Patients frequently avoided consulting dermatologists due to time constraints or because they see others using the same medications for similar skin issues [39]. Research in Australia revealed that individuals experiencing skin conditions reported increased self-consciousness, heightened embarrassment, and feelings of social exclusion [40]. A study in Nepal found that the lack of awareness and the shortage of dermatologists, especially in rural areas, led to increased OTC medication use [27]. Similarly, our study observed that individuals often went directly to drug dispensers because of the unawareness about the presence of dermatologist within their reach or sometimes simply because of lack of knowledge on who to consult.

In our study, most people when asked on why they did not visit specialist care, seemed to be unaware of the concept and benefits of health insurance. Similarly, data from the Government of Nepal in 2021 showed that approximately 29% of families were unaware of the health insurance policy, and 34.1% were aware but did not enroll [41]. Additionally, limited healthcare coverage and high rates of infectious diseases in Nepal [42,43], led many to rely on OTC medication [44,45]. Pharmacies/drug shops were popular due to their accessibility, low cost, and convenience, despite offering lower quality care. These medical shops were preferred because they involved minimal direct, indirect, and opportunity costs, making them an attractive option for many individuals [8]. The OTC use of medications emerges as a result of systemic limitations compounded by patients' treatment-seeking behaviour [45,46]. For instance, at the health system level, specialist care for skin conditions is limited, particularly in terms of human resources, and this is most pronounced in peripheral areas of the country [21–25,47,48]. At the same time, treatment-seeking behaviour is influenced by the perceived severity of skin conditions and the convenience of accessing informal health services [46,49,50]. As a result, widespread OTC purchases of medicines, including topical corticosteroids, reflect a complex interplay of regulatory gaps, limited healthcare access, and convenience [8,45]. Addressing this issue requires a multi-pronged approach that targets both systemic health services limitations and public engagement.

The abundance of online information posed a risk, as patients often couldn't distinguish between credible and unreliable sources. For example, a teenager with acne might find home remedies such as applying apple cider vinegar or

toothpaste, leading to burns [51]. This issue echoes with our study, where participants reported using unconventional and potentially harmful remedies such as goat's urine and *Harpic* (a toilet cleanser solution) for their skin conditions. Skin diseases were often underestimated with attitudes that were expressed as *"It's just a rash"* or *"it's not life-threatening"*. However, only recently has there been recognition of the substantial impact that chronic, disfiguring, or disabling skin diseases had on the quality of life of patients and their families [52]. Additionally, there was a common practice of reusing old prescriptions for recurrent or new lesions [53], which also echoes with our participants.

The research highlights the significant problem of steroid misuse and evading treatment seeking at the skin specialists which has been echoed across many countries such as Iraq, Nepal and India where a substantial percentage of dermatology outpatients were improperly using TCs [20,54,55]. In India, misuse of TCs on the face has become a major concern, leading to conditions like steroid addiction or red face syndrome. This results in severe rebound erythema, burning sensations, and scaling when TCs use is discontinued after long-term use [11].

In our study, participants reported experiencing similar side effects after using TCs, such as a burning sensation, redness, and thinning of the skin. Concerning side effects, dosage, and proper usage instructions, 97.9% of patients in India were unaware of the potential adverse effects of TCs and had not read the manufacturer's leaflet [53]. In Saudi Arabia, research showed low awareness of corticosteroid side effects among both users and non-users, with users slightly more informed [56]. Similarly, participants in our study were unaware of steroids and had never read the medication composition even when they were found to use it for a long time.

## Implications

The study highlights the high OTC use of topical corticosteroid and thus warrants an urgent set of efforts to curb the treatment seeking behaviour. First, community and public engagement efforts are critical to enhance the awareness of skin conditions and the rationale for consulting physicians which could safeguard the ethics and interests of various stakeholders [57,58]. Public awareness activities should focus on the fundamentals of skin conditions, consequences, and the use of corticosteroids, including the concomitant use of antimicrobials, could be streamlined as a concerted effort. The potential consequences of delayed treatment seeking behaviour [46,49,50] as outlined in this study, further highlight the addition of both direct and indirect costs, stress, time and resources on both supplier and provider sides. Future studies could explore these dimensions through health economics research, including stakeholder analysis (e.g., examining how stakeholders' interests interact and influence policy and practice), to further investigate the potential for policy changes. Future studies could integrate quantitative data collection and analysis to complement the qualitative findings, including exploring the magnitude of OTC steroid use for skin conditions. In addition, the OTC use of these products is highly likely to trigger the concomitant use of antimicrobials and thus antimicrobial resistance interventions would require a concerted effort. Third, engagement with dispensers, pharmacies (drug shops) and stakeholders are critical to promote their understanding of the OTC use of steroids, beauty products and antimicrobials and their consequences. Multi-pronged engagement on the basics of various medications (e.g., steroids, antimicrobials), their over-the-counter use, and associated implications should be promoted through both formal and informal media. Public engagement initiatives—such as podcasts and dialogues [59] in accessible language—are especially critical. Regulatory scrutiny on the widespread use of topical corticosteroids, including advertisements in the media on beauty products such as recommending their categories and extent, is critical. The International League of Dermatological Societies (ILDS), an umbrella organization for professional dermatological societies, has released a position paper with two key recommendations [60] that align with this study's findings:

• Moderate to strong topical corticosteroids, whether alone or combined with other active compounds like antimicrobials, should only be used when prescribed by a properly trained health professional.

• In countries where unregulated use is common, clear guidelines should be in place to protect patients and the public from overuse or misuse.

The ILDS recommendations, along with the World Health Assembly's recognition of skin health as a global public health priority for universal health coverage in May 2025, highlight the urgent need to strengthen primary health care in low- and middle-income countries [61].

**Strengths and limitations of study**

The study included specific respondents with skin conditions and a history of using TCs for a variable period, thus it may not address the treatment seeking behaviour for other general skin conditions. These respondents had rich lived experiences and thus the insights provided by these respondents highlight the scenario of skin conditions among patients including factors influencing topical steroid misuse. Nonetheless, by virtue of selecting these specific patients we may have incurred selection bias. Furthermore, as this is a clinic-based study only those cases with symptoms secondary to steroid abuse were recruited, potentially overlooking asymptomatic instances of steroid misuse. The use of a qualitative method with a specific cohort of participants from a specialist clinic limits the generalizability of the findings to broader social and cultural contexts.

**Conclusions**

Lack of knowledge about skin diseases, perceived lack of severity and reliance on friends, family and home remedies at the initial stage seem to have delayed treatment. Lack of recovery and development of complications often prompted these patients to seek OTC medications; and when these medications failed to cure their skin conditions, patients sought treatment at the formal health services including dermatologists. The barriers in reaching formal health services, specialist clinic and most importantly, patients' disposition to underestimate skin conditions appeared to have affected the outcome of these skin conditions. An early treatment seeking at formal health services (specialist care) is critical to ensure that these patients save resources and receive appropriate treatment before the development of consequences. A multifaceted approach embedded in public and community engagement is essential to address the treatment-seeking behaviors of patients with skin conditions.

**Supporting information**

**S1_Text. COREQ: COnsolidated criteria for REporting Qualitative research.**
(PDF)

**S2_Text. Interview guide adapted for the study.**
(DOCX)

**S3_Text. Inclusivity in Global Health.**
(DOCX)

**S1_Fig. Coding Framework.**
(PDF)

**Acknowledgments**

We are grateful to the respondents who generously provided additional time when visiting clinics and insights for this study.

**Author contributions**

**Conceptualization:** Yogesh Poudyal, Anuja Rajbhandari, Mukti Ghimire, Sanjaya Acharya, Shristi Raut, Sunil Pokharel, Phaik Yeong Cheah, Bipin Adhikari.

**Data curation:** Yogesh Poudyal, Nishika Aryal, Anuja Rajbhandari.

**Formal analysis:** Yogesh Poudyal, Nishika Aryal, Anuja Rajbhandari, Bipin Adhikari.

**Funding acquisition:** Yogesh Poudyal, Bipin Adhikari.

**Investigation:** Yogesh Poudyal, Bipin Adhikari.

**Methodology:** Yogesh Poudyal, Nishika Aryal, Anuja Rajbhandari, Mukti Ghimire, Sanjaya Acharya, Shristi Raut, Bishal Joshi, Sunil Pokharel, Phaik Yeong Cheah, Bipin Adhikari.

**Project administration:** Yogesh Poudyal.

**Resources:** Yogesh Poudyal, Bipin Adhikari.

**Software:** Yogesh Poudyal, Nishika Aryal.

**Supervision:** Yogesh Poudyal, Mukti Ghimire, Sanjaya Acharya, Shristi Raut, Sunil Pokharel, Phaik Yeong Cheah, Bipin Adhikari.

**Validation:** Yogesh Poudyal, Nishika Aryal, Bipin Adhikari.

**Visualization:** Bipin Adhikari.

**Writing – original draft:** Yogesh Poudyal, Nishika Aryal, Anuja Rajbhandari, Bipin Adhikari.

**Writing – review & editing:** Yogesh Poudyal, Mukti Ghimire, Sanjaya Acharya, Shristi Raut, Bishal Joshi, Sunil Pokharel, Phaik Yeong Cheah, Bipin Adhikari.

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
