## [Decision Letter · Decision Letter 0]

PGPH-D-24-02509

Over the counter use of topical corticosteroid for skin conditions among patients before attending skin specialist clinic in Nepal: a qualitative study.

Dear Dr. Poudyal,

Thank you for submitting your manuscript to PLOS Global Public Health. After careful consideration, we feel that it has merit but does not fully meet PLOS Global Public Health’s publication criteria as it currently stands. Therefore, we invite you to submit a revised version of the manuscript that addresses the points raised during the review process.

The manuscript has been evaluated by three reviewers, and their comments are available below.

The reviewers have raised a number of concerns that need attention. In particular, they request additional information on methodological aspects of the study, and further discussion.

Could you please revise the manuscript to carefully address the concerns raised?

We look forward to receiving your revised manuscript.

Kind regards,

Helen Howard

Staff Editor

Journal Requirements:

Additional Editor Comments (if provided):

Reviewers' comments:

Reviewer's Responses to Questions

**Comments to the Author**

1. Does this manuscript meet PLOS Global Public Health’s publication criteria ? Is the manuscript technically sound, and do the data support the conclusions? The manuscript must describe methodologically and ethically rigorous research with conclusions that are appropriately drawn based on the data presented.

Reviewer #1: Yes

Reviewer #2: Yes

Reviewer #3: Yes

2. Has the statistical analysis been performed appropriately and rigorously?

Reviewer #1: Yes

Reviewer #2: N/A

Reviewer #3: N/A

3. Have the authors made all data underlying the findings in their manuscript fully available (please refer to the Data Availability Statement at the start of the manuscript PDF file)?

Reviewer #1: Yes

Reviewer #2: Yes

Reviewer #3: Yes

4. Is the manuscript presented in an intelligible fashion and written in standard English?

Reviewer #1: Yes

Reviewer #2: Yes

Reviewer #3: Yes

5. Review Comments to the Author

Reviewer #1: 1. Methods: Study period (Sept 2021 to March 2022: Was there any rationale for the period of the study, as around 40 skin specific patients were present in the clinic per day and your sample size (N=31)?

2. Clearly specify the inclusion and exclusion criteria of the study subjects.

3. Respondent Selection (L10): Was the dispensers (pharmacist) attitude towards TCs, OTC dispensing verified?

4. Results (Broad Themes & Consolidation of Themes):

a + b = 1. Skin conditions and the knowledge gaps;

c = 2. Treatment seeking behavior

d, where dis you consolidate d (perception) as 'c ' corresponds to practice and not perception?

5. Table 1: (Age) You have conducted a face to face interview with socio-demographic details. Mention the exact age and NOT in range.. Example 1) 48 years, 2) 26 years etc.

6. Discussion: Was there any discussion done with the pharmacists (drug store or pharmacies} to understand the trend for self medication. Did'nt the pharmacist refer them to specialist clinics? What is the basis of the statement?

7. P16, para 2: "These medical shops were preferred because they involved minimal direct, indirect, and opportunity costs, making them an attractive option for many individuals." - Was there any discussion done with the pharmacists (drug store or pharmacies} to understand the trend for self medication. Did'nt the pharmacist refer them to specialist clinics. What is the basis of the statement?

8. Reference: Please check the journal referencing guidelines and edit all the references accordingly.

Reviewer #2: PGPH-D-24-02509

Review steroid

Abstract: Conclusion: suggest switching the word “insurmountable” to something more optimistic and achievable eg difficult or tough as want to indicate potential solution not just a problem that cannot be fixed.

Introduction

Suggest emphasizing that dermatophytosis as a superficial fungal infection is in the top 4 of all disease burdens in GBD study (Vos T., Abajobir A.A., Abate K.H. Global, regional, and national incidence, prevalence, and years lived with disability for 328 diseases and injuries for 195 countries, 1990-2016: a systematic analysis for the Global Burden of Disease Study 2016. Lancet. 2017;390(10100):1211–1259. doi: 10.1016/S0140-6736(17)32154-2) This is the hook to bring in the general reader.

Also suggest including a brief explanation of the access to dermatology care options in Nepal, Do all patient have to pay is there free care available via a public system? You discuss this a bit in the discussion section but I think it needs summarising in the introduction and study context.

Would be good to highlight the global context of this issue possibly referencing the ILDS/AAD platform "Dermatology Covid-19, Monkeypox, and Emerging Infections Registry" https://redcap.partners.org/redcap/surveys/index.php?s=YJWAJCX7TY

Method

Study setting

You indicate the study was done in a clinic which I assume is private although not clear. How much do the patient pay to access? Where else could they go? Is there dermatology services available in the local tertiary hospital you mention,

You indicate you recruit over 6 months when the clinic would have seen about 5000 patients and yet you interview/include just 31 patients (less than 1%) – Given the premise of the study that steroid OTC is a common problem in your setting, can you indicate how you selected the cases? Presumably you did not include all patients who admitted to using OTC steroid as this must be higher than <1%. I think it would help the manuscript to understand more about the case selection process and what about those who you identified but did not take part would strengthen the validity of the conclusions you then draw

In terms of conclusions

Referencing in the International League of Dermatologists Position Statement on the safe and appropriate use of topical steroids might strengthen your message as will calling for an increase in including access to skin health as a global public health priority and essential for universal health coverage alluding to the recently proposed and put forward as an agenda item for World Health Assembly in May 2025

See concept note @ https://globalskin.org/doclink/skin-diseases-as-a-global-public-health-priority/eyJ0eXAiOiJKV1QiLCJhbGciOiJIUzI1NiJ9.eyJzdWIiOiJza2luLWRpc2Vhc2VzLWFzLWEtZ2xvYmFsLXB1YmxpYy1oZWFsdGgtcHJpb3JpdHkiLCJpYXQiOjE3MTY0OTc4NzksImV4cCI6MTcxNjU4NDI3OX0.1VKfoy3Nd_ZJMr-JI0uCV1cSSI41uCjrDjSrbdMmNk0#:~:text=NTD%202021%20%2D2030%20roadmap%20and,of%20resources%2C%20policy%20and%20advocacy.&text=previous%20unmet%20need%2C%20expand%20access,target%20to%20deliver%20on%20UHC.

I hope these comments are helpful

Reviewer #3: This article is a very important in the current scenario of topical steroid abuse in our country. It is good to see article related to topical steroid misuse being published from public health point but it would have been better if the authors could also do analysis of the responses.

6. PLOS authors have the option to publish the peer review history of their article (what does this mean? ). If published, this will include your full peer review and any attached files.

**Do you want your identity to be public for this peer review?** For information about this choice, including consent withdrawal, please see our Privacy Policy .

Reviewer #1: **Yes: ** Abdul Nazer Ali

Reviewer #2: **Yes: ** Dr Lucinda Claire Fuller

Reviewer #3: No

---

## [Decision Letter · Decision Letter 1]

PGPH-D-24-02509R1

Over the counter use of topical corticosteroid for skin conditions among patients before attending skin specialist clinic in Nepal: a qualitative study.

Dear Dr. Poudyal,

Thank you for submitting your manuscript to PLOS Global Public Health. After careful consideration, we feel that it has merit but does not fully meet PLOS Global Public Health’s publication criteria as it currently stands. Therefore, we invite you to submit a revised version of the manuscript that addresses the points raised during the review process.

The manuscript has been evaluated by two reviewers, and their comments are available below.

The reviewers have raised a number of concerns. Could you please carefully revise the manuscript to address all comments raised? In your response to the reviewers, I would recommend that you clearly state the changes you have made to your manuscript so that the reviewer can easily evaluate your revisions. Particularly, I would recommend that you include again the detailed response to Reviewer 1s feedback from the previous rounds, in addition to your comments regarding the latest feedback, as those revisions had been missed. 

We look forward to receiving your revised manuscript.

Kind regards,

Johanna Pruller, Ph.D.

PLOS Staff Editor

Journal Requirements:

Additional Editor Comments (if provided):

Reviewers' comments:

Reviewer's Responses to Questions

**Comments to the Author**

1. If the authors have adequately addressed your comments raised in a previous round of review and you feel that this manuscript is now acceptable for publication, you may indicate that here to bypass the “Comments to the Author” section, enter your conflict of interest statement in the “Confidential to Editor” section, and submit your "Accept" recommendation.

Reviewer #1: (No Response)

Reviewer #2: (No Response)

2. Does this manuscript meet PLOS Global Public Health’s publication criteria ? Is the manuscript technically sound, and do the data support the conclusions? The manuscript must describe methodologically and ethically rigorous research with conclusions that are appropriately drawn based on the data presented.

Reviewer #1: Partly

Reviewer #2: Yes

3. Has the statistical analysis been performed appropriately and rigorously?

Reviewer #1: N/A

Reviewer #2: N/A

4. Have the authors made all data underlying the findings in their manuscript fully available (please refer to the Data Availability Statement at the start of the manuscript PDF file)?

Reviewer #1: No

Reviewer #2: Yes

5. Is the manuscript presented in an intelligible fashion and written in standard English?

Reviewer #1: Yes

Reviewer #2: Yes

6. Review Comments to the Author

Reviewer #1: No comments raised in a previous round (15. 11 2025) of review has been addresses nor justified. It is just the same manuscript sent wasting the reviewers time. However, Having reviewed again, I register the following comments,

The article addresses an important and under-researched topic: the over-the-counter (OTC) use of topical corticosteroids (TCS) in Nepal, particularly among patients before they seek specialized dermatological care. The study employs a qualitative approach, which is appropriate for exploring the motivations, perceptions, and experiences of patients regarding self-medication with TCS. The findings have significant implications for public health, dermatological practice, and policy-making in low-resource settings like Nepal. However, the manuscript requires revisions to strengthen its methodological rigor, clarity, and overall impact.

Areas for Improvement:

1. Methodological Clarity:

- The sampling strategy is not clearly described. How were participants selected? What criteria were applied in purposive sampling?

- The interview guide should be included as a supplementary file or described in detail to ensure transparency and reproducibility.

- The process of data saturation is mentioned but not adequately explained. How was saturation determined?

2. Data Analysis:

- The article lacks detail on the qualitative data analysis process. Was a specific framework (e.g., thematic analysis, grounded theory) used? How were themes identified and validated? Including a coding tree or thematic framework as a supplementary file would enhance transparency.

- The role of the researchers in the analysis process (e.g., reflexivity, potential biases) is not discussed. This is important for establishing the credibility of the findings.

3. Findings and Interpretation:

- The results section is descriptive but could benefit from deeper interpretation. For example, what do the findings imply about the cultural or socio-economic factors driving OTC use of TCS in Nepal?

- The article does not sufficiently contextualize the findings within the broader literature on OTC medication use in LMICs. How do the results compare with similar studies in other regions?

4. Ethical Considerations:

- Ethical approval and informed consent are mentioned, but more details are needed. Were participants informed about the potential risks of discussing sensitive topics? How was confidentiality maintained?

5. Writing and Structure:

- The introduction could be more concise and focused. The background information on TCS and their misuse is relevant but could be streamlined to better highlight the research gap.

- The discussion section should be expanded to include recommendations for policy and practice. For example, how can healthcare providers and policymakers address the issue of OTC TCS use in Nepal?

6. Limitations:

- The limitations of the study are not adequately addressed. For instance, the generalizability of the findings to other regions or populations is limited due to the qualitative design and specific context of Nepal. This should be explicitly acknowledged.

Reviewer #2: Thanks for this opportunity to review this revised manuscript on an important topic.

I hope the comments are helpful

Abstract Introduction

Suggest you change the word “use” to “access” or “availability” The current sentence doesn’t make sense.

Introduction:

The reference supporting your opening statement that steroids have emerged as the primary choice for dermatologists in the outpatient settings is a review of the use of steroids and does not include source data to support this claim. The statement is made in the reference with no supporting source.

Suggest you rephrase to say TCs are widely prescribed by the dermatologist…….etc

Das A, Panda S. Use of topical corticosteroids in dermatology: an evidence-based approach. Indian journal of dermatology. 2017;62(3):237-50

Might you prepare a summary of the participants in terms of male/female, and number in each age range, with the actual age of the participants on your full table of demographics.

7. PLOS authors have the option to publish the peer review history of their article (what does this mean? ). If published, this will include your full peer review and any attached files.

**Do you want your identity to be public for this peer review?** For information about this choice, including consent withdrawal, please see our Privacy Policy .

Reviewer #1: **Yes: ** Dr. Abdul Nazer Ali

Reviewer #2: **Yes: ** Dr L Claire Fuller

---

## [Decision Letter · Decision Letter 2]

PGPH-D-24-02509R2

Over the counter use of topical corticosteroid for skin conditions among patients before attending skin specialist clinic in Nepal: a qualitative study.

Dear Dr. Poudyal,

Thank you for submitting your manuscript to PLOS Global Public Health. After careful consideration, we feel that it has merit but does not fully meet PLOS Global Public Health’s publication criteria as it currently stands. Therefore, we invite you to submit a revised version of the manuscript that addresses the points raised during the review process.

Your revised manuscript has been reassessed by reviewers 1 and 2. Reviewer 1 has some minor revisions that we would like you to address in your revised manuscript please, in addition to providing a point-by-point response. 

We look forward to receiving your revised manuscript.

Kind regards,

Sarah Jose, Ph.D.

Staff Editor

Journal Requirements:

Additional Editor Comments (if provided):

Reviewers' comments:

Reviewer's Responses to Questions

**Comments to the Author**

1. If the authors have adequately addressed your comments raised in a previous round of review and you feel that this manuscript is now acceptable for publication, you may indicate that here to bypass the “Comments to the Author” section, enter your conflict of interest statement in the “Confidential to Editor” section, and submit your "Accept" recommendation.

Reviewer #1: All comments have been addressed

Reviewer #2: All comments have been addressed

2. Does this manuscript meet PLOS Global Public Health’s publication criteria ? Is the manuscript technically sound, and do the data support the conclusions? The manuscript must describe methodologically and ethically rigorous research with conclusions that are appropriately drawn based on the data presented.

Reviewer #1: Yes

Reviewer #2: Yes

3. Has the statistical analysis been performed appropriately and rigorously?

Reviewer #1: N/A

Reviewer #2: Yes

4. Have the authors made all data underlying the findings in their manuscript fully available (please refer to the Data Availability Statement at the start of the manuscript PDF file)?

Reviewer #1: No

Reviewer #2: No

5. Is the manuscript presented in an intelligible fashion and written in standard English?

Reviewer #1: Yes

Reviewer #2: Yes

6. Review Comments to the Author

Reviewer #1: 1. Review the manuscript for minor grammatical errors and typographical issues to improve clarity and flow.

2. If feasible, provide a coding framework or thematic map as supplementary material to enhance transparency of the analysis process.

3. Expand on systemic factors in discussion (e.g., pharmacy practices, regulation gaps, healthcare access) contributing to OTC corticosteroid use.

4. Provide more concrete and actionable public health recommendations (e.g., public awareness campaigns, regulatory enforcement)

Reviewer #2: The ILDS recommendations, along with the World Health Assembly's recognition of skin health as a

global public health priority for universal health coverage in May 2024, highlight the urgent need to

strengthen primary health care in low- and middle-income countries [59]

It is actually on the agenda for May 2025 (not 2024)

7. PLOS authors have the option to publish the peer review history of their article (what does this mean? ). If published, this will include your full peer review and any attached files.

**Do you want your identity to be public for this peer review?** For information about this choice, including consent withdrawal, please see our Privacy Policy .

Reviewer #1: No

Reviewer #2: **Yes: ** Dr Lucinda Claire Fuller

---

## [Editor Report · Decision Letter 3]

Over the counter use of topical corticosteroid for skin conditions among patients before attending skin specialist clinic in Nepal: a qualitative study.

PGPH-D-24-02509R3

Dear Dr Poudyal,

We are pleased to inform you that your manuscript 'Over the counter use of topical corticosteroid for skin conditions among patients before attending skin specialist clinic in Nepal: a qualitative study.' has been provisionally accepted for publication in PLOS Global Public Health.

Best regards,

Julia Robinson

Executive Editor